# Lower Macromolecular Content in Tendons of Female Patients with Osteoporosis versus Patients with Osteopenia Detected by Ultrashort Echo Time (UTE) MRI

**DOI:** 10.3390/diagnostics12051061

**Published:** 2022-04-24

**Authors:** Saeed Jerban, Yajun Ma, Amir Masoud Afsahi, Alecio Lombardi, Zhao Wei, Meghan Shen, Mei Wu, Nicole Le, Douglas G. Chang, Christine B. Chung, Jiang Du, Eric Y. Chang

**Affiliations:** 1Department of Radiology, University of California, San Diego, CA 92093, USA; yam013@health.ucsd.edu (Y.M.); amasoudafsahi@health.ucsd.edu (A.M.A.); allombardi@health.ucsd.edu (A.L.); z2wei@health.ucsd.edu (Z.W.); mshen@ucsd.edu (M.S.); mew140@health.ucsd.edu (M.W.); nile@health.ucsd.edu (N.L.); cbchung@health.ucsd.edu (C.B.C.); jiangdu@health.ucsd.edu (J.D.); 2Research Service, Veterans Affairs San Diego Healthcare System, San Diego, CA 92161, USA; 3Department of Orthopaedic Surgery, University of California, San Diego, CA 92093, USA; dchang@health.ucsd.edu

**Keywords:** tendon, MRI, ultrashort TE, magnetization transfer, aging, macromolecular proton

## Abstract

Tendons and bones comprise a special interacting unit where mechanical, biochemical, and metabolic interplays are continuously in effect. Bone loss in osteoporosis (OPo) and its earlier stage disease, osteopenia (OPe), may be coupled with a reduction in tendon quality. Noninvasive means for quantitatively evaluating tendon quality during disease progression may be critically important for the improvement of characterization and treatment optimization in patients with bone mineral density disorders. Though clinical magnetic resonance imaging (MRI) sequences are not typically capable of directly visualizing tendons, ultrashort echo time MRI (UTE-MRI) is able to acquire a high signal from tendons. Magnetization transfer (MT) modeling combined with UTE-MRI (i.e., UTE-MT-modeling) can indirectly assess macromolecular proton content in tendons. This study aimed to determine whether UTE-MT-modeling could detect differences in tendon quality across a spectrum of bone health. The lower legs of 14 OPe (72 ± 6 years) and 31 OPo (73 ± 6 years) female patients, as well as 30 female participants with normal bone (Normal-Bone, 36 ± 19 years), are imaged using UTE sequences on a 3T MRI scanner. Institutional review board approval is obtained for the study, and all recruited subjects provided written informed consent. A T1 measurement and UTE-MT-modeling are performed on the anterior tibialis tendon (ATT), posterior tibialis tendon (PTT), and the proximal Achilles tendon (PAT) of all subjects. The macromolecular fraction (MMF) is estimated as the main measure from UTE-MT-modeling. The mean MMF in all the investigated tendons was significantly lower in OPo patients compared with the Normal-Bone cohort (mean difference of 24.2%, *p* < 0.01), with the largest Normal-Bone vs. OPo difference observed in the ATT (mean difference of 32.1%, *p* < 0.01). Average MMF values of all the studied tendons are significantly lower in the OPo cohort compared with the OPe cohort (mean difference 16.8%, *p* = 0.02). Only the PPT shows significantly higher T1 values in OPo patients compared with the Normal-Bone cohort (mean difference 17.6%, *p* < 0.01). Considering the differences between OPo and OPe groups with similar age ranges, tendon deterioration associated with declining bone health was found to be larger than a priori detected differences caused purely by aging, highlighting UTE-MT MRI techniques as useful methods in assessing tendon quality over the course of progressive bone weakening.

## 1. Introduction

The tendon and bone are complementary tissues that comprise a special interacting unit that is essential for locomotion and individual autonomy [1]. With mechanical interplays continuously in effect, bone acts as a lever for skeletal muscle, and tendons connects bone mechanically to muscle, facilitating the effective transfer of force. The internal mechanical forces applied to these tissues are generated by muscle contractions that affect the strength and microarchitecture of both bone [1,2] and tendon [3]. For example, Chen et al. have shown that the tendon’s mechanical strength is significantly correlated with the bone mineral density in an osteoporotic rabbit model [4]. In addition to mechanical interactions, there are biochemical and metabolic dynamics between bone and tendon that have significant impacts on their quality and function. Anabolic hormone levels and inflammatory cytokine activity (e.g., growth hormone (GH), insulin-like growth factor-I (IGF-I), and sex steroids), in addition to the anabolic or catabolic molecules secreted by tendon or bone tissues, can affect cell activity in both of them [5,6,7,8].

The bone loss in osteoporosis (OPo) and in its earlier stage disease, osteopenia (OPe), may be coupled with a reduction in tendon quality. Using quantitative noninvasive methods to evaluate the quality of tendon during disease progression may be of critical interest should the diagnosis and treatment of OPe and OPo improve.

Though magnetic resonance imaging (MRI), a noninvasive imaging modality, is routinely used for the morphological evaluation of tendons [9,10,11], clinical MRI techniques face significant challenges in the detection and clinical evaluation of tendons [12,13,14]. Tendons possess very short T2 relaxation times due to a high concentration of an organized collagenous matrix [14] that results in a low signal-to-noise ratio when imaged using conventional MR sequences, obfuscating the accurate differentiation of healthy tendon from abnormal tissue.

In contrast to conventional MRI sequences, ultrashort echo time (UTE) MRI techniques are capable of detecting a considerable signal from tendons [14]. Consequently, UTE-MRI is able to provide robust MRI-based biomarkers useful for the quantitative assessment of tendons, such as apparent relaxation time of transverse magnetization (T2*), the relaxation time of longitudinal magnetization (T1), and magnetization transfer (MT) measurements such as MT ratio (MTR) [12,13,14,15,16,17,18,19,20,21].

Another critical barrier to the use of clinical MRI and UTE-MRI for the quantitative assessment of tendons is the confounding factor of the so-called “magic angle effect.” This phenomenon affects collagen-rich tissues like tendons [22] when the tissue fiber orientation alters from 0° to 55° relative to the main magnetic field axis, B_0_, and the MR properties can increase by over 200% [23,24,25,26,27].

The MT modeling combined with UTE-MRI has recently been used to indirectly measure macromolecular proton fraction (MMF) in different tissues [25,26,28,29,30,31,32,33,34,35]. In theory, the off-resonance MR pulses saturate macromolecular proton magnetization, which transfers to the water pool and is subsequently measured on the UTE MR images in order to estimate the ratio between protons in macromolecules and those in water pools (i.e., MMF). In tendons, the dominant macromolecule is collagen, composing 60–85% of the tendons’ dry weight [36,37]. Interestingly, two-pool UTE-MT modeling has demonstrated promise as a clinically compatible quantitative technique that is resistant to the magic angle effect [25,26]. UTE-MT modeling was recently implemented on cadaveric Achilles and rotator cuff tendons to assess MMF and showed roughly no orientation angle sensitivity [25,26]. This technique provides multiple parameters, including MMF, macromolecular relaxation time (T2mm), and exchange rates [25,26].

Earlier in vivo investigations on human tibial tendons demonstrated age-related decreases in MMF in the anterior tibialis tendon (ATT) and posterior tibialis tendon (PTT) [38] by comparing young and elderly female cohorts, but it is still unclear how sensitive MT parameters may be to the changes in tendons across a spectrum of bone health. Should UTE-MT modeling demonstrate significant sensitivity to tendon quality, namely, its deterioration, over the course of osteoporosis development, the technique could play a potential role in improving the characterization and optimizing treatment in these patients.

This study aimed to investigate the differences in MMF of lower leg tendons between female osteopenia (OPe) patients, osteoporosis (OPo) patients, and participants with normal bone (Normal-Bone). By considering OPe and OPo patients with a similar range of age, the differences between the OPe and OPo groups can be explained relative to the bone quality differences.

## 2. Material and Methods

### 2.1. Participants’ Recruitment

Fourteen OPe (72 ± 6 years) and thirty-one OPo (73 ± 6 years) female patients, as well as thirty female participants with normal bone (Normal-Bone, 36 ± 19 years), were recruited by flyer to undergo leg MRI scans. The inclusion criteria for each cohort were as follows: (1) Normal-Bone cohort: healthy pre-menopausal females under 40 years of age or post-menopausal females with recent (<one month) DEXA T-scores above −1; (2) OPe cohort: post-menopausal females with DEXA T-scores between −2.5 and −1; (3) OPo cohort: post-menopausal females with DEXA T-scores below −2.5. The institutional review board (IRB) of the University of California, San Diego approved this study, which was conducted in accordance with applicable good clinical practice requirements and the relevant guidelines and regulations. Written informed consent was obtained from each participant.

### 2.2. UTE-MRI Scanning

Participants were asked to decide on their own which leg was to be scanned, regardless of leg dominance. Their lower leg was imaged using UTE-MRI sequences on a 3T MRI (MR750, GE Healthcare Technologies, WI, USA) with an eight-channel knee coil for both radiofrequency (RF) transmission and signal reception. The imaging slab was centered at the middle of the shin and localized based on operator experience. Field of view (FOV), matrix dimension, nominal in-plane pixel size, and slice thickness were 14 cm, 160 × 160, 0.87 mm, and 5 mm, respectively.

To measure UTE-T_1_ as a prerequisite for the two-pool UTE-MT modeling, an actual flip angle-variable repetition time (AFI-VTR)-based 3D-UTE-Cones sequence (AFI: TE = 0.032 ms; TRs = 20 ms and 100 ms; flip angle (FA) = 45°; VTR: TE = 0.032 ms; repetition times (TRs) = 20, 80, and 150 ms; FA = 45°; rectangular RF pulse with a duration of 150 µs) was performed with a total scan time of 20 min [39]. T_1_ was measured based on a single-component exponential fitting (S(TR)∝1−exp(−TR/T1)+constant) of the acquired data [39].

Additionally, a 3D-UTE-Cones-MT sequence (Fermi saturation pulse power levels = 500°, 1000°, and 1500°; frequency offset = 2, 5, 10, 20, and 50 kHz; FA = 7°; 9 spokes per MT preparation; rectangular RF excitation pulse of 100 µs) was performed for two-pool MT modeling with a total scan time of 13 min [40,41,42]. Using two-pool UTE-MT modeling, MMF was calculated as an index quantifying the ratio of the macromolecular proton pool over the water pool. For UTE-MT modeling, the acquired data with the set of MT saturation pulse power levels were fitted by a previously described modified rectangular pulse approximation approach [25,28,43]. A super-Lorentzian lineshape function was used to model the macromolecular proton spectrum and the loss of the longitudinal magnetization of the macromolecular pool [28].

The UTE-T1 and UTE-MT analyses were performed offline on the acquired DICOM images using Matlab (version 2017, Mathworks, Natick, MA, USA) codes written in-house. A Levenberg–Marquardt algorithm was employed for non-linear least-squares fitting in both the UTE-MT modeling and T1 fitting within the selected regions of interest (ROIs). These fitting models were then used to generate pixel maps of MMF.

Figure 1A,B show representative axial images of the lower leg of a healthy 25-year-old female participant using clinical gradient echo (GRE) and UTE-MRI sequences, respectively.

### 2.3. Data Analysis

Elastix software (Version 4.9.0, Stefan Kelin, Rotterdam, Netherland, open-source software: http://elastix.isi.uu.nl/, accessed on 1 October 2018) was used to register all images to the first T1 image (TR = 20 ms) to compensate for potential subject motion. All scans were smoothed using a Gaussian filter with a 3 × 3 sub-window before T1 and MT measurements.

To investigate the reproducibility of the MRI data analysis, ROIs were selected by three experienced readers at the ATT, PTT, and proximal Achilles tendon (PAT). Figure 1C shows a schematic of the selected ROIs in a representative MRI image from a 25-year-old female subject (TR = 80 ms and TE = 2 ms). For quality control purposes, the ATT, PTT, and PAT ROIs selected by the readers were overseen and validated by a board-certified musculoskeletal (MSK) radiologist for the first five datasets. Intraclass correlation coefficients (ICCs) were measured for T1 and MMF comparisons across all estimations made by the three independent radiology readers.

### 2.4. Statistical Analysis

The one-sample Kolmogorov–Smirnov test determined that the measured UTE-MRI parameters were not normally distributed, so the Kruskal–Wallis test by ranks was used to examine the differences in tendon quality between the three participant cohorts (OPe, OPo, and Normal-Bone). As an exploratory investigation, UTE-MRI parameters were compared between the studied tendons (ATT, PTT, and PAT) within each recruited group. The *p*-values below 0.05 were considered as significant. The Holm-Bonferroni method was used to correct the significance level for multiple comparisons and all UTE-MRI measurements and statistical analyses were performed using Matlab (version 2017, The Mathworks Inc., Natick, MA, USA) codes developed in-house.

## 3. Results

Figure 2 shows MMF maps generated for the tendons of three exemplary participants from the Normal-Bone, OPe, and OPo cohorts. For these examples, MMF values were observed in the following ascending order: OPo < OPe < Normal-Bone.

Table 1 presents the estimated average and standard deviation (SD) values of T1 and MMF values in the ATTs, PTTs, and PATs for the Normal-Bone, OPe, and OPo cohorts. ICCs between the three independent measurements are also presented in Table 1 for T1 and MMF values. For all MRI parameters, ICC was higher than 0.95, which indicates a high level of consistency between the measurements performed by independent readers.

The percentage differences and statistical significances (Kruskal–Wallis test by ranks, corrected for multiple comparisons) of T1 and MMF between the Normal-Bone, OPe, and OPo cohorts are presented in Table 2. The MRI parameters investigated in tendon did not show significant differences between the Normal-Bone and OPe cohorts. Remarkably, MMF values in all investigated tendons were significantly lower in OPo patients compared with the Normal-Bone cohort (>24.2%), with the largest Normal-Bone vs. OPo difference in the ATT (32.1%). Though MMF values measured for each tendon separately did not show significant differences between the OPe and OPo cohorts, but the average MMF value over all the studied tendons was significantly lower in OPo patients compared with the OPe cohort (16.8%). T1 was significantly higher in OPo patients compared with the Normal-Bone cohort only for the PPT (17.6%).

Table 3 presents the percentage differences and statistical significance of T1 and MMF between the different tendons that were studied. The T1 of ATT was higher than the PTT in all groups, while the difference was significant only in the Normal-Bone group. Remarkably, the T1 of ATT was significantly higher than the PAT for all groups. The T1 of PTT was higher than the PAT in all groups, while the difference was significant only in the OPo group. MMF was lower in ATT compared with PTT as well as in PAT compared with PTT in all groups, while the differences were significant only for the Normal-Bone group. The MMF differences between ATT and PAT were not significant.

Figure 3 depicts the average, median, SD, and the first and third quartiles of the T1 and MMF values for each cohort using box and whisker plots. Statistically significant differences are indicated with horizontal red lines between cohorts and highlighted with an asterisk.

## 4. Discussion

The MMF obtained from two-pool UTE-MT modeling applied to 3D-UTE-Cones data was employed to detect bone mineral density-related differences in lower leg tendon quality as indicated by estimated differences in the macromolecule content. In prior studies, UTE-MT modeling was shown to be an orientation-insensitive MRI technique [26,28] useful for noninvasive assessment of musculoskeletal tissue quality [31,35], with detected differences likely predicated on changes in the macromolecule fraction of the investigated tissues. A recent study demonstrated age-related changes in tibialis tendons, where MMF was 16.8% to 23.0% lower in an elderly group (75 ± 8 years old) compared with young healthy participants (29 ± 6 years old). It should be noted that the previously demonstrated differences may be partially explained by the menopause condition affecting only the elderly group. In the current study, MMF in ATTs, PTTs, and PATs was found to be significantly lower for OPo patients compared with a Normal-Bone cohort, with differences ranging from 24.2% to 32.1%, which are larger than those previously reported due to advancing age. Interestingly, the average MMF values over all studied tendons were lower in OPo patients compared with OPe patients (16.8%). Due to the similar range of ages in the OPe and OPo groups, their tendon differences can be explained by their bone quality differences.

In general, aging affects the quality, composition, and performance of tendons [44,45,46,47]. Aging can directly impact tendon biomechanics, causing biological changes that ultimately result in weaker tendons [44,45,47]. Age-induced reductions in tendon stiffness, a crucial measure of function, have been reported to be anywhere between 2% and 55%, depending on the tendon studied [45,46]. Decreasing mechanical loads that result from a combination of weakening muscles and lowered levels of physical activity can also cause adaptations in tendon microstructure and composition [45,47]. Age-related reductions in muscle strength of up to 52% have been reported in the literature [45]. Age-related changes in the cross-section area of tendons have also been investigated, but the reported variation patterns across different studies have been inconsistent [45,46,48].

Simultaneous with age-related changes, tendons can be affected by episodes of osteoporosis-related bone weakening. Specifically, reduced levels of growth hormone (GH), insulin-like growth factor-I (IGF-I), and sex steroids—commonly recognized pathways to bone and muscle deterioration [5,6,7,8,49,50]—are able to produce remarkable reductions in protein turnover and cell activity in bone, muscle, and tendon [1,3,5,51]. For example, the tendon’s mechanical strength has been shown to be significantly correlated to the bone mineral density in an osteoporotic rabbit model [4]. In the current study, the changes in tendons induced by osteoporosis, as opposed to aging, were highlighted by the detected lower average MMF values in the tendons of OPo patients compared with OPe patients with a similar average age (Table 2).

Hodgson et al. [52] used a 2D UTE version of MT modeling and reported lower MMF values in the Achilles tendons of patients with psoriatic arthritis (~16%) compared with those of healthy volunteers (~21%). Ma et al. used the 2D UTE-MT technique on Achilles tendon specimens and demonstrated the orientation-insensitivity of the UTE-MT modeling approach, reporting mean MMF values of 20% [25]. Zhu et al. later used 3D Cones UTE-MT modeling to assess rotator cuff tendon specimens and reported significantly lower MMF values for specimens of severe tendinopathy compared with specimens of mild tendinopathy (16% vs. 12%, respectively) [26]. While the MMF values of ATT and PTT in our study are within the range of these previously reported values [25,26], it is important to note that tendon composition, collagenous matrix morphology, and biomechanical properties vary not only between tendons but in a region-dependent manner within the same tendon [13,18,53].

Comparing the UTE-MRI parameters between the studied tendons showed significantly lower MMF in ATT compared with PTT as well as in PAT compared with PTT in all groups, while the differences were significant only for the Normal-Bone group. Remarkably, these differences were smaller for the OPo and OPe groups (i.e., the elderly).

The first limitation of this study was the relatively long scan time (approximately 34 min), which may have made it difficult for subjects to remain motionless during the scan. The motion was generally not substantial on the images obtained from most subjects, but image registration helped in the few cases where motion occurred between series. Employing acceleration techniques, such as stretching the readout trajectory, could be used to further accelerate the technique and limit scan time to 20 min without producing significant errors [54]. Moreover, using a reasonable constant value for T1 depending on the age of the participants in order to exclude T1 measurement (~20 min of scan time) could be feasible for shortening scan time. A future in vivo study will be performed to generate a selection chart for reasonable T1 values as a function of age. Secondly, this study limited its subject pool to female participants as a means to avoid misinterpretation of osteoporosis-related tendon variations and the sex-related differences [55,56]. Performing a similar study on an exclusively male patient population and comparing the results with this study’s findings on female patients will be necessary in the future. Thirdly, this investigation only focused on lower leg tendons, where pathology is quite common [9,37,57,58,59]. However, other tendons, such as the patellar tendon, are of comparable clinical significance and should also be evaluated using this study’s proposed MRI techniques in a future investigation. Fourthly, participants were asked to decide on their own which leg was to be scanned. It is expected that bone deterioration progresses simultaneously on both sides, though there may be variations between tendons on the left and right sides, possibly related to leg dominance. Future investigations should be performed to study the tendon and bone quality differences between legs in healthy controls as well as in patients. Fifthly, the observed UTE-MRI measured differences between the Normal-Bone and the patient groups may be partially explained by the large age differences between the groups as well as the difference in their menopause conditions. Future investigations should be performed by including an age-matched, post-menopausal healthy female group (DEXA T-score > −1) in order to exclude the potential influences of age and the menopause condition on the final conclusions.

## 5. Conclusions

Two-pool UTE-MT modeling was employed to assess osteoporosis-related changes in lower leg tendon quality. The MMF obtained from the UTE-MT modeling, used as an index for macromolecule content in tendons, showed significantly lower values in the lower leg tendons of the OPo cohort compared with the OPe and Normal-Bone cohort. Due to the similar ranges of age in the OPe and OPo groups, their detected tendon differences can be explained relative to the bone quality differences. Tendon deterioration associated with osteoporotic development was found to be larger than a priori detected differences caused purely by aging, highlighting the UTE-MT MR technique as a useful method in assessing tendon quality over the course of progressive bone weakening during the development of osteoporosis.

## Figures and Tables

**Figure 1 diagnostics-12-01061-f001:**
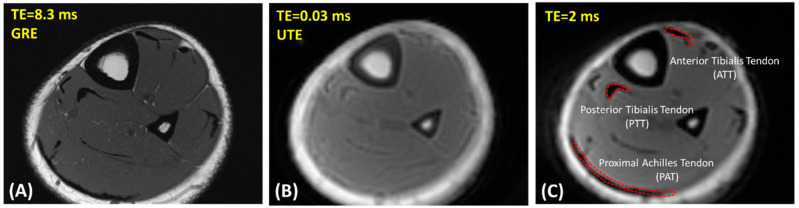
Representative axial images of the lower leg of a healthy 25-year-old female participant using (**A**) clinical gradient echo (GRE) sequence (TR = 790 ms, TE = 8.3 ms, in-plane matrix = 352 × 352), (**B**) ultrashort echo time (UTE) Cones MRI sequence (TR = 80 ms, TE = 0.032 ms, in-plane matrix = 160 × 160), and (**C**) Cones MRI sequence at TE = 2 ms (TR = 80 ms, in-plane matrix = 160 × 160). Representative regions of interest (ROIs) for anterior and posterior tibialis (ATT and PTT) tendons and proximal Achilles (PAT) tendons were selected at TE = 2 ms because it provided higher contrast (indicated by red dashed boundary).

**Figure 2 diagnostics-12-01061-f002:**
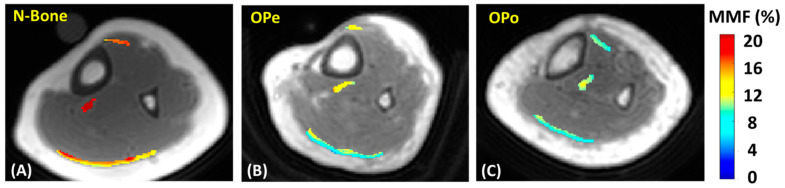
Generated macromolecular proton fraction (MMF) maps for exemplary participants from (**A**) Normal-Bone cohort (23-year-old female), (**B**) OPe cohort (83-year-old female), and (**C**) OPo cohort (85-year-old female). MMF was higher in Normal-Bone participant compared with OPe and OPo patients. N-Bone label refers to Normal-Bone group.

**Figure 3 diagnostics-12-01061-f003:**
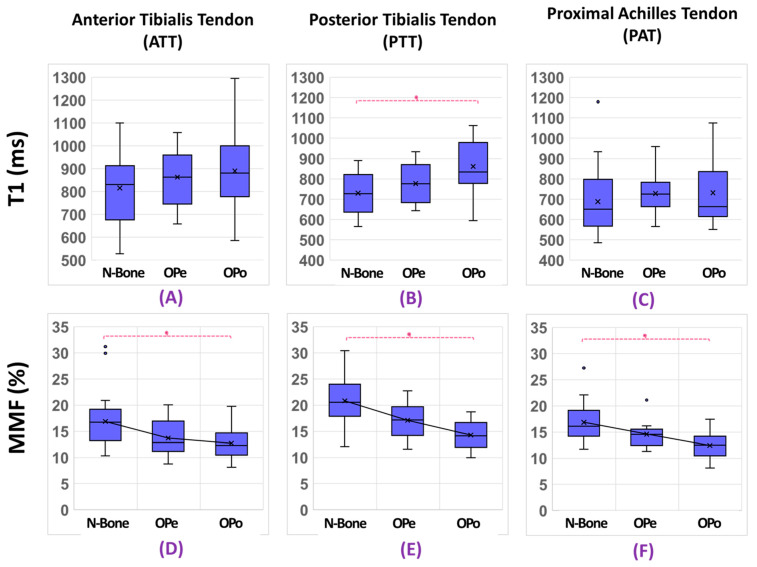
Average UTE-T1 and MMF in (**A**,**D**) ATT, (**B**,**E**) PTT, and (**C**,**F**) PAT for Normal-Bone, OPe, and OPo cohorts. The central mark in each plot indicates the median, while the bottom and top edges of the box indicate the 25th and 75th percentiles, respectively. MMF was significantly lower in OPo patients compared with Normal-Bone cohort for all tendons. T1 was significantly higher in OPo patients compared with the Normal-Bone cohort only for PPT. N-Bone label refers to Normal-Bone group.

**Table 1 diagnostics-12-01061-t001:** Mean and standard deviation of T1 and UTE-MT measurements in ATT, PTT, and PAT for Normal-Bone, OPe, and OPo cohorts. Intraclass correlation coefficient (ICC) was also measured between the three independent readers. N-Bone label refers to Normal-Bone group.

	T1 (ms)				MMF (%)			
	ATT	PTT	PAT	Mean	ATT	PTT	PAT	Mean
**N-Bone**	829 ± 145	733 ± 91	698 ± 156	758 ± 86	16.8 ± 4.7	21.0 ± 4.2	16.8 ± 3.5	18.2 ± 3.6
**OPe**	832 ± 144	766 ± 101	702 ± 111	757 ± 99	14.3 ± 3.6	17.2 ± 3.1	15.0 ± 2.9	15.8 ± 3.6
**OPo**	890 ± 180	861 ± 127	732 ± 164	825 ± 145	12.7 ± 2.7	14.3 ± 2.6	12.4 ± 2.5	13.2 ± 2.5
**ICC**	0.98 ± 0.01	0.97 ± 0.02	0.96 ± 0.02	0.98 ± 0.01	0.98 ± 0.01	0.99 ± 0.01	0.97 ± 0.01	0.99 ± 0.01

**Table 2 diagnostics-12-01061-t002:** Average percentage differences as well as Kruskal–Wallis test results of T1 and MMF between Normal-Bone, OPe, and OPo cohorts. N-Bone label refers to Normal-Bone group.

	T1 Difference (%)				MMF Difference (%)			
	ATT	PTT	PAT	Mean	ATT	PTT	PAT	Mean
**N-Bone Vs. OPe**	0.3 (*p* = 1.00)	4.5 (*p* = 0.70)	0.7 (*p* = 0.79)	−0.1 (*p* = 0.98)	−18.3 (*p* = 0.03)	−14.6 (*p* = 0.17)	−10.7 (*p* = 0.28)	−12.9 (*p* = 0.17)
**N-Bone Vs. OPo**	7.3 (*p* = 0.47)	**17.6** **(*p* < 0.01)**	4.9 (*p* = 0.67)	8.7 (*p* = 0.31)	**−32.1** **(*p* < 0.01)**	**−24.2** **(*p* < 0.01)**	**−26.1** **(*p* < 0.01)**	**−27.5** **(*p* < 0.01)**
**OPe Vs. OPo**	6.9 (*p* = 0.65)	12.5 (*p* = 0.06)	4.2 (*p* = 1.00)	8.9 (*p* = 0.32)	−16.9 (*p* = 0.07)	−11.3 (*p* = 0.30)	−17.3 (*p* = 0.04)	**−16.8** **(*p* = 0.02)**

**Table 3 diagnostics-12-01061-t003:** Average percentage differences as well as Kruskal–Wallis test results of T1 and MMF between ATT, PTT, and PAT tendons. N-Bone label refers to Normal-Bone group.

	T1 Difference (%)			MMF Difference (%)		
	ATT-PTT	ATT-PAT	PTT-PAT	ATT-PTT	ATT-PAT	PTT-PAT
**N-Bone**	−11.67 (*p* = 0.05)	**−15.90** **(*p* < 0.01)**	−4.78 (*p* = 0.34)	**25.43** **(*p* < 0.01)**	0.45 (*p* = 0.93)	**−19.92** **(*p* < 0.01)**
**OPe**	−7.95 (*p* = 0.4)	**−15.61** **(*p* = 0.02)**	−8.32 (*p* = 0.38)	19.94 (*p* = 0.04)	5.04 (*p* = 0.92)	−12.42 (*p* = 0.10)
**OPo**	−3.16 (*p* = 0.95)	**−17.75** **(*p* < 0.01)**	**−15.06** **(*p* < 0.01)**	12.38 (*p* = 0.06)	−2.10 (*p* = 0.95)	−12.88 (*p* = 0.03)

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
