# Peer review of "Lower Macromolecular Content in Tendons of Female Patients with Osteoporosis versus Patients with Osteopenia Detected by Ultrashort Echo Time (UTE) MRI"

_diagnostics, 2022, doi:10.3390/diagnostics12051061_

Round 1

Reviewer 1 Report

The authors propose an original article exploring the role of UTE MRI in assessing changes in tendon of patients with osteopenia and osteoporosis compared to healthy controls.

The methodology is well deisigned, as well as presentation of results.

However, the introduction is too long, and authors should better clarify the research question (e.g. previous reports of tendinopathy associated with osteoporosis? preclinical strudies of common pathophysiological pathways?)

Author Response

Reviewer #1:

We greatly appreciate your comments and suggestions which significantly improved our manuscript. Please find our responses to your comments below.

Please note that the manuscript has been revised accordingly and that changes are highlighted in yellow. We have indicated sections in the manuscript which refer directly to reviewer comments with notations in the margin (e.g., R1. Comment 1).

Comments

However, the introduction is too long, and authors should better clarify the research question (e.g. previous reports of tendinopathy associated with osteoporosis? preclinical studies of common pathophysiological pathways?)

Response: We appreciate the comment and the suggestions. We were not able to find many studies investigating the tendinopathy associated with osteoporosis literature. Chen et al., have shown that the tendon’s mechanical strength is significantly correlated with the bone mineral density in an osteoporotic rabbit model (Chen et al., 2015). Tendons can be affected by episodes of osteoporosis-related bone weakening. Specifically, reduced levels of growth hormone (GH), insulin-like growth factor-I (IGF-I), and sex steroids—commonly recognized pathways to bone and muscle deterioration (Adams, 2001; Girgis et al., 2014; Giustina et al., 2008; Reginster et al., 2016; Tagliaferri et al., 2015; Urban, 2011)—are able to produce remarkable reductions in protein turnover and cell activity in bone, muscle, and tendon (Boesen et al., 2014; Edwards et al., 2015; Frizziero et al., 2014; Tagliaferri et al., 2015). This information has been added to the Introduction section for clarification. The introduction section has been shortened as suggested.

References

Adams, G.R., 2001. Insulin-like growth factor in muscle growth and its potential abuse by athletes. West. J. Med. 175, 7–9. doi:10.1136/ewjm.175.1.7

Boesen, A.P., Dideriksen, K., Couppé, C., Magnusson, S.P., Schjerling, P., Boesen, M., Aagaard, P., Kjaer, M., Langberg, H., 2014. Effect of growth hormone on aging connective tissue in muscle and tendon: Gene expression, morphology, and function following immobilization and rehabilitation. J. Appl. Physiol. 116, 192–203. doi:10.1152/japplphysiol.01077.2013

Chen, X., Giambini, H., Ben-Abraham, E., An, K.N., Nassr, A., Zhao, C., 2015. Effect of bone mineral density on rotator cuff tear: An osteoporotic rabbit model. PLoS One 10, 1–12. doi:10.1371/journal.pone.0139384

Edwards, M.H., Dennison, E.M., Aihie Sayer, A., Fielding, R., Cooper, C., 2015. Osteoporosis and sarcopenia in older age. Bone 80, 126–130. doi:10.1016/j.bone.2015.04.016

Frizziero, A., Vittadini, F., Gasparre, G., Masiero, S., 2014. Impact of oestrogen deficiency and aging on tendon: Concise review. Muscles. Ligaments Tendons J. 4, 324–328. doi:10.11138/mltj/2014.4.3.324

Girgis, C.M., Mokbel, N., DiGirolamo, D.J., 2014. Therapies for musculoskeletal disease: Can we treat two birds with one stone? Curr. Osteoporos. Rep. doi:10.1007/s11914-014-0204-5

Giustina, A., Mazziotti, G., Canalis, E., 2008. Growth hormone, insulin-like growth factors, and the skeleton. Endocr. Rev. doi:10.1210/er.2007-0036

Reginster, J.Y., Beaudart, C., Buckinx, F., Bruyère, O., 2016. Osteoporosis and sarcopenia: Two diseases or one? Curr. Opin. Clin. Nutr. Metab. Care 19, 31–36. doi:10.1097/MCO.0000000000000230

Tagliaferri, C., Wittrant, Y., Davicco, M.J., Walrand, S., Coxam, V., 2015. Muscle and bone, two interconnected tissues. Ageing Res. Rev. doi:10.1016/j.arr.2015.03.002

Urban, R.J., 2011. Growth hormone and testosterone: Anabolic effects on muscle. Horm. Res. Paediatr. 76, 81–83. doi:10.1159/000329184

Reviewer 2 Report

The manuscript „Decreased collagen content in tendons of patients with osteoporosis and osteopenia detected with ultrashort echo time 3 (UTE) MRI” presents the next results obtained using UTE-based magnetic resonance imaging sequences by the group from the University of California, San Diego, CA.  

Unfortunately, the authors included two controversial terms in the manuscript title: „collagen content” and „patients with osteoporosis and osteopenia”. In my opinion, both terms are unjustified, and it is the main drawback of the study.

1.

In the introduction, the authors stated that „MMF in a tendon is likely an indicator of estimated collagen content …”. Also in the discussion, one can find a passage “… detected differences likely predicated on changes in the collagen fraction of the investigated tissues”. Concerning „collagen content” as used in the manuscript title, there is no explicit statement in the manuscript that collagen content equals MMF (or how exactly it is related to MMF). There are no results denoted as „collagen content” in tables and charts (Table 1, Table 2 and Figure 3)! Please be precise.

Referencing their previous study, the authors stated in the introduction, that “it is still unclear how sensitive MT parameters may be to the compositional and structural changes of tendon across a spectrum of bone health.” In my opinion, these issues are not touched in the presented manuscript.

2.

The title and aim of the study (lines 105-107) suggest that the investigation concerns healthy subjects and patients (subjects with OPe and OPo) – not age dependence of the „collagen content”. But mean age of the control cohort was 36 years and of patients was 72 or 73 (roughly two times older). The authors included pre-menopausal females under 40 into the control cohort and it lowered the mean age in this cohort. Maybe the differences in MMF between control and Opo/Ope cohorts are partially age-dependent?

Additionally, it is well known, that females undergo extensive menopause-induced changes. OPe and OPo cohorts were exclusively post-menopausal females, while control did not. Can the differences in MMF between control and Opo/Ope cohorts be partially induced by menopausal transitions?

What about exclusively post-menopausal healthy females as the control cohort?

Fortunately, „the changes in tendons induced by osteoporosis, as opposed to aging, were highlighted by the detected lower average MMF values in tendons of OPo patients compared with OPe patients with similar average age” (lines 248-251).

Moreover, the results cannot be generalized in the title as „patients with osteoporosis and osteopenia” as these patients were exclusively female patients with all consequences derived from this fact! The authors stated in the discussion that „Secondly, this study limited its subject pool to female participants as a means to avoid misinterpretation of osteoporosis-related tendon variations and the sex-related differences [54,55]. Performing a similar study on an exclusively male patient population and comparing the results with this study’s findings on female patients will be necessary in the future.” It further confirms that the study cannot be generalized to all patients (the title suggests sex independence). Moreover, in my opinion, starting with more complicated case (i.e., female patients) was not a good idea.

Please, make the narration consistent in this aspects. 

3.

What about similarities/differences between particular tendons? Have the authors tried to explore this aspect? Looking at the results (tables and figures), it seems that there is space for some discussions.

4.

Some minor issues:

What is the „reasonable constant value for T1” (line 270)?

What is the meaning of „desirable leg to be scanned” (line 121). Does it mean dominant leg? The authors assumed that there is no difference between legs?

Please, unify the fonts in Table 1 (the last row is different).

Author Response

Reviewer #2:

We greatly appreciate your comments and suggestions which significantly improved our manuscript. Please find our responses to your comments below.

Please note that the manuscript has been revised accordingly and that changes are highlighted in yellow. We have indicated sections in the manuscript which refer directly to reviewer comments with notations in the margin (e.g., R1. Comment 10).

The manuscript „Decreased collagen content in tendons of patients with osteoporosis and osteopenia detected with ultrashort echo time 3 (UTE) MRI” presents the next results obtained using UTE-based magnetic resonance imaging sequences by the group from the University of California, San Diego, CA.  

Unfortunately, the authors included two controversial terms in the manuscript title: „collagen content” and „patients with osteoporosis and osteopenia”. In my opinion, both terms are unjustified, and it is the main drawback of the study.

Response: We appreciate the comment and the suggestions. As described in following responses, the manuscript’s title has been modified based on the reviewer’s suggestions.

  1. In the introduction, the authors stated that „MMF in a tendon is likely an indicator of estimated collagen content …”. Also in the discussion, one can find a passage “… detected differences likely predicated on changes in the collagen fraction of the investigated tissues”. Concerning „collagen content” as used in the manuscript title, there is no explicit statement in the manuscript that collagen content equals MMF (or how exactly it is related to MMF). There are no results denoted as „collagen content” in tables and charts (Table 1, Table 2 and Figure 3)! Please be precise.

Response: We appreciate the comment and the suggestions. The reviewer is correct that the UTE-MT sequence measures macromolecules more broadly. The theory is that the off-resonance MR pulses saturate macromolecular proton magnetization, which transfers to the water pool and is subsequently measured on the UTE MR images. The MMF is the ratio between macromolecular and water pool protons. In tendon, the dominant macromolecule is collagen, composing 60-85% of the tendons’ dry weight (Aparecida de Aro et al., 2012; Khan et al., 1999). As suggested by the reviewer, the manuscript and its title has been modified to mainly refer to MMF instead of collagen.

Referencing their previous study, the authors stated in the introduction, that “it is still unclear how sensitive MT parameters may be to the compositional and structural changes of tendon across a spectrum of bone health.” In my opinion, these issues are not touched in the presented manuscript.

Response: We appreciate the comment and agree with the reviewer. Since the composition and structure of the tendons were not directly examined in this study, we have modified the sentence as follows:

“it is still unclear how sensitive MT parameters may be to the changes of tendon across a spectrum of bone health”.

  1. The title and aim of the study (lines 105-107) suggest that the investigation concerns healthy subjects and patients (subjects with OPe and OPo) – not age dependence of the „collagen content”. But mean age of the control cohort was 36 years and of patients was 72 or 73 (roughly two times older). The authors included pre-menopausal females under 40 into the control cohort and it lowered the mean age in this cohort. Maybe the differences in MMF between control and Opo/Ope cohorts are partially age-dependent?

Additionally, it is well known, that females undergo extensive menopause-induced changes. OPe and OPo cohorts were exclusively post-menopausal females, while control did not. Can the differences in MMF between control and Opo/Ope cohorts be partially induced by menopausal transitions?

What about exclusively post-menopausal healthy females as the control cohort?

Fortunately, „the changes in tendons induced by osteoporosis, as opposed to aging, were highlighted by the detected lower average MMF values in tendons of OPo patients compared with OPe patients with similar average age” (lines 248-251).

Moreover, the results cannot be generalized in the title as „patients with osteoporosis and osteopenia” as these patients were exclusively female patients with all consequences derived from this fact! The authors stated in the discussion that „Secondly, this study limited its subject pool to female participants as a means to avoid misinterpretation of osteoporosis-related tendon variations and the sex-related differences [54,55]. Performing a similar study on an exclusively male patient population and comparing the results with this study’s findings on female patients will be necessary in the future.” It further confirms that the study cannot be generalized to all patients (the title suggests sex independence). Moreover, in my opinion, starting with more complicated case (i.e., female patients) was not a good idea.

Please, make the narration consistent in this aspects. 

Response: We appreciate the comment and the suggestions. Please note that in our earlier study we have demonstrated the age-related decreases of MMF lower leg tendons (Jerban et al., 2019) by comparing young and elderly female cohorts. In the previous study MMF was 16.8% to 23.0% lower in the elderly group (75±8 years old) compared with young healthy participants (29±6 years old). As suggested, these changes are not only a result of the age differences but also caused by the menopause condition. The following sentence has been added to the first paragraph of the Discussion section for clarification:

It should be noted that the previously demonstrated differences may be partially explained by the menopause condition affecting only the elderly group.

The main novelty of the current study was investigating the tendon differences between female OPe (72±6 years) and OPo (73±6 years) groups with similar ranges of age. Therefore, the differences between OPe and OPo groups can be explained relative to the bone quality differences. Interestingly, the average MMF values over all studied tendons were lower in OPo patients compared with OPe patients (16.8%).

As suggested, the manuscript and its title have been modified to clarify the focus of this study on the female OPe-OPo differences

Title: Lower macromolecular content in tendons of female patients with osteoporosis versus patients with osteopenia detected by ultrashort echo time (UTE) MRI

  1. What about similarities/differences between particular tendons? Have the authors tried to explore this aspect? Looking at the results (tables and figures), it seems that there is space for some discussions.

Response: We appreciate the suggestion. A new exploratory investigation has been added to the manuscript comparing T1 and MMF of the studied tendons. Briefly,

Table 3 presents the percentage differences and statistical significances of T1 and MMF between the different tendons that were studied. T1 of ATT was higher than PTT in all groups, while the difference was significant only in the Ctrl group. Remarkably, T1 of ATT was significantly higher than PAT for all groups. T1 of PTT was higher than PAT in all groups, while the difference was significant only in OPo group. MMF was lower in ATT compared with PTT as well as in PAT compared with PTT in all groups, while the differences were significant only for Ctrl group. MMF differences between ATT and PAT were not significant.

  1. Some minor issues:
    • What is the „reasonable constant value for T1” (line 270)?

Response: We appreciate the comment. We plan to decrease the scan time from approximately 35 minutes to less than 15 minutes in future studies. We hope to only focus on the UTE-MT sequences that are the core of our study for macromolecular fraction (an indication of collagen content) estimation and exclude the T1 measurement for each participant. We have suggested performing a separate future in vivo study to generate a selection chart for reasonable T1 values as a function of age to be used as the input in our UTE-MT modeling. This has been clarified in the last paragraph of the Discussion section as follows:

Moreover, using a reasonable constant value for T1 depending on the age of the participants in order to exclude T1 measurement (~20 minutes of scan time) could be feasible for shortening scan time. A future in vivo study will be performed to generate a selection chart for reasonable T1 values as a function of age.

  • What is the meaning of „desirable leg to be scanned” (line 121). Does it mean dominant leg? The authors assumed that there is no difference between legs?

Response: We appreciate the comment. Participants were asked to decide on their own which leg was to be scanned. It is expected that bone deterioration progresses simultaneously on both sides, though we acknowledge that there may be variations between tendons on the left and right sides, possibly related to leg dominance. We have clarified this in the methods and added the following to the limitations section:

Fourthly, participants were asked to decide on their own which leg was to be scanned. It is expected that bone deterioration progresses simultaneously on both sides, though there may be variations between tendons on the left and right sides, possibly related to leg dominance. Future investigations should be performed to study the tendon and bone quality differences between legs in healthy controls as well as in patients.

  • Please, unify the fonts in Table 1 (the last row is different).

Response: We appreciate the comment. The font size has been modified as suggested.

References

Aparecida de Aro, A., de Campos Vidal, B., Pimentel, E.R., 2012. Biochemical and anisotropical properties of tendons. Micron 43, 205–214. doi:10.1016/j.micron.2011.07.015

Jerban, S., Ma, Y., Namiranian, B., Ashir, A., Shirazian, H., Zhao, W., Wu, M., Cai, Z., Le, N., Du, J., Chang, E.Y.E.Y., Wei, Z., Le, N., Wu, M., Cai, Z., Du, J., Chang, E.Y.E.Y., 2019. Age-related decrease in collagen proton fraction in tibial tendons estimated by magnetization transfer modeling of ultrashort echo time magnetic resonance imaging (UTE-MRI). Sci. Rep. November, 17974. doi:10.1038/s41598-019-54559-3

Khan, K.M., Cook, J.L., Bonar, F., Harcourt, P., Astrom, M., 1999. Histopathology of common tendinopathies: update and implications for clinical management. Sport. Med. 27, 393–408. doi:10.2165/00007256-199927060-00004

Round 2

Reviewer 1 Report

Authors satisfactorily addressed my comments

Author Response

We highly appreciate your support.

Reviewer 2 Report

The authors provided reasonable answers to issues raised in my review and modified relevant parts of the manuscript. But there is one exception. The authors have not addressed one important issue, and I suspect they tried to avoid touching it.  According to my knowledge, the control cohort should include individuals similar to the studied (patients) cohort(s) in terms of age, gender, race, area of residence, and other factors. As I have pointed out in my original review, the control cohort is younger (36 vs 72/73) and heterogeneous regarding age and state (pre- vs. post-menopausal) :

“The title and aim of the study (lines 105-107) suggest that the investigation concerns healthy subjects and patients (subjects with OPe and OPo) – not age dependence of the „collagen content”. But mean age of the control cohort was 36 years and of patients was 72 or 73 (roughly two times older). The authors included pre-menopausal females under 40 into the control cohort and it lowered the mean age in this cohort. Maybe the differences in MMF between control and Opo/Ope cohorts are partially age-dependent?

Additionally, it is well known, that females undergo extensive menopause-induced changes. OPe and OPo cohorts were exclusively post-menopausal females, while control did not. Can the differences in MMF between control and Opo/Ope cohorts be partially induced by menopausal transitions?

What about exclusively post-menopausal healthy females, as the control cohort?”

As in the previous sentence, my suggestion is post-menopausal healthy females, preferably at ~70, as the control. Please improve or provide a strong rationale regarding assumptions the authors made when selecting individuals for the control cohort (such rationale, if provided, should be included in the revised version of the manuscript).

Author Response

We greatly appreciate your comments and suggestions. Please find our responses to your comments in the attached response letter.

Round 3

Reviewer 2 Report

The explanations and subsequent changes in the manuscript are satisfactory. It seems that, at the moment, conclusions are based on the results presented.

The manuscript can be accepted in its current form.

This manuscript is a resubmission of an earlier submission. The following is a list of the peer review reports and author responses from that submission.